# Improved Network Robustness
# with Adversary Critic

**Alexander Matyasko, Lap-Pui Chau**
School of Electrical and Electronic Engineering
Nanyang Technological University, Singapore
aliaksan001@ntu.edu.sg, elpchau@ntu.edu.sg

## Abstract

Ideally, what confuses neural network should be confusing to humans. However, recent experiments have shown that small, imperceptible perturbations can change the network prediction. To address this gap in perception, we propose a novel approach for learning robust classifier. Our main idea is: adversarial examples for the robust classifier should be indistinguishable from the regular data of the adversarial target. We formulate a problem of learning robust classifier in the framework of Generative Adversarial Networks (GAN), where the adversarial attack on classifier acts as a generator, and the critic network learns to distinguish between regular and adversarial images. The classifier cost is augmented with the objective that its adversarial examples should confuse the adversary critic. To improve the stability of the adversarial mapping, we introduce adversarial cycle-consistency constraint which ensures that the adversarial mapping of the adversarial examples is close to the original. In the experiments, we show the effectiveness of our defense. Our method surpasses in terms of robustness networks trained with adversarial training. Additionally, we verify in the experiments with human annotators on MTurk that adversarial examples are indeed visually confusing.

## 1  Introduction

Deep neural networks are powerful representation learning models which achieve near-human performance in image [1] and speech [2] recognition tasks. Yet, state-of-the-art networks are sensitive to small input perturbations. [3] showed that adding *adversarial noise* to inputs produces images which are visually similar to the original inputs but which the network misclassifies with high confidence. In speech recognition, [4] introduced an adversarial attack, which can change any audio waveform, such that the corrupted signal is over $99.9\%$ similar to the original but transcribes to any targeted phrase. The existence of *adversarial examples* puts into question generalization ability of deep neural networks, reduces model interpretability, and limits applications of deep learning in safety and security-critical environments [5, 6].

Adversarial training [7, 8, 9] is the most popular approach to improve network robustness. Adversarial examples are generated online using the latest snapshot of the network parameters. The generated adversarial examples are used to augment training dataset. Then, the classifier is trained on the mixture of the original and the adversarial images. In this way, adversarial training smoothens a decision boundary in the vicinity of the training examples. Adversarial training (AT) is an intuitive and effective defense, but it has some limitations. AT is based on the assumption that adversarial noise is label non-changing. If the perturbation is too large, the adversarial noise may change the true underlying label of the input. Secondly, adversarial training discards the dependency between the model parameters and the adversarial noise. As a result, the neural network may fail to anticipate changes in the adversary and overfit the adversary used during training.

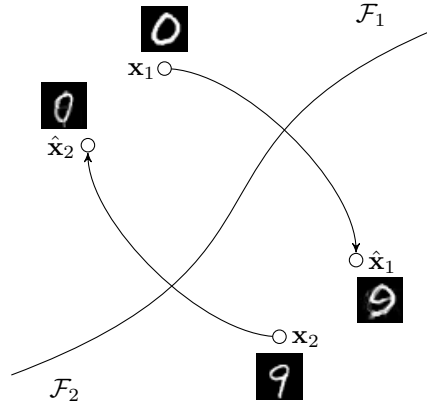

Figure 1: Adversarial examples should be indistinguishable from the regular data of the adversarial target. The images in the figure above are generated using Carlini and Wagner [10] $l_2$-attack on the network trained with our defense, such that the confidence of the prediction on the adversarial images is 95%. The confidence on the original images $x_1$ and $x_2$ is 99%.

Ideally, what confuses neural network should be confusing to humans. So the changes introduced by the adversarial noise should be associated with removing identifying characteristics of the original label and adding identifying characteristics of the adversarial label. For example, images that are adversarial to the classifier should be visually confusing to a human observer. Current techniques [7, 8, 9] improve robustness to input perturbations from a selected uncertainty set. Yet, the model's adversarial examples remain semantically meaningless. To address this gap in perception, we propose a novel approach for learning robust classifier. Our core idea is that adversarial examples for the robust classifier should be indistinguishable from the regular data of the attack's target class (see fig. 1).

We formulate the problem of learning robust classifier in the framework of Generative Adversarial Networks (GAN) [11]. The adversarial attack on the classifier acts as a generator, and the critic network learns to distinguish between natural and adversarial images. We also introduce a novel targeted adversarial attack which we use as the generator. The classifier cost is augmented with the objective that its adversarial images generated by the attack should confuse the adversary critic. The attack is fully-differentiable and implicitly depends on the classifier parameters. We train the classifier and the adversary critic jointly with backpropagation. To improve the stability of the adversarial mapping, we introduce adversarial cycle-consistency constraint which ensures that the adversarial mapping of the adversarial examples is close to the original. Unlike adversarial training, our method does not require adversarial noise to be label non-changing. To the contrary, we require that the changes introduced by adversarial noise should change the "true" label of the input to confuse the critic. In the experiments, we demonstrate the effectiveness of the proposed approach. Our method surpasses in terms of robustness networks trained with adversarial training. Additionally, we verify in the experiments with human annotators that adversarial examples are indeed visually confusing.

## 2   Related work

**Adversarial attacks**   Szegedy et al. [3] have originally introduced a targeted adversarial attack which generates adversarial noise by optimizing the likelihood of input for some adversarial target using a box-constrained L-BFGS method. Fast Gradient Sign method (FGSM) [7] is a one-step attack which uses a first-order approximation of the likelihood loss. Basic Iterative Method (BIM), which is also known as Projected Gradient Descent (PGD), [12] iteratively applies the first-order approximation and projects the perturbation after each step. [6] propose an iterative method which at each iteration selects a single most salient pixel and perturbs it. DeepFool [13] iteratively generates adversarial perturbation by taking a step in the direction of the closest decision boundary. The decision boundary is approximated with first-order Taylor series to avoid complex non-convex optimization. Then, the geometric margin can be computed in the closed-form. Carlini and Wagner [10] propose an optimization-based attack on a modified loss function with implicit box-constraints. [14] introduce a black-box adversarial attack based on transferability of adversarial examples. Adversarial Transformation Networks (ATN) [15] trains a neural network to attack.

**Defenses against adversarial attacks** Adversarial training (AT) [7] augments training batch with adversarial examples which are generated online using Fast Gradient Sign method. Virtual Adversarial training (VAT) [16] minimizes Kullback-Leibler divergence between the predictive distribution of clean inputs and adversarial inputs. Notably, adversarial examples can be generated without using label information and VAT was successfully applied in semi-supervised settings. [17] applies iterative Projected Gradient Descent (PGD) attack to adversarial training. Stability training [18] minimizes a task-specific distance between the output on clean and the output on corrupted inputs. However, only a random noise was used to distort the input. [19, 20] propose to maximize a geometric margin to improve classifier robustness. Parseval networks [21] are trained with the regularization constraint, so the weight matrices have a small spectral radius. Most of the existing defenses are based on robust optimization and improve the robustness to perturbations from a selected uncertainty set.

Detecting adversarial examples is an alternative way to mitigate the problem of adversarial examples at test time. [22] propose to train a detector network on the hidden layer's representation of the guarded model. If the detector finds an adversarial input, an autonomous operation can be stopped and human intervention can be requested. [23] adopt a Bayesian interpretation of Dropout to extract confidence intervals during testing. Then, the optimal threshold was selected to distinguish natural images from adversarial. Nonetheless, Carlini and Wagner [24] have extensively studied and demonstrated the limitations of the detection-based methods. Using modified adversarial attacks, such defenses can be broken in both white-box and black-box setups. In our work, the adversary critic is somewhat similar to the adversary detector. But, unlike adversary-detection methods, we use information from the adversary critic to improve the robustness of the guarded model during training and do not use the adversary critic during testing.

**Generative Adversarial Networks** [11] introduce a generative model where the learning problem is formulated as an adversarial game between discriminator and generator. The discriminator is trained to distinguish between real images and generated images. The generator is trained to produce naturally looking images which confuse the discriminator. A two-player minimax game is solved by alternatively optimizing two models. Recently several defenses have been proposed which use GAN framework to improve robustness of neural networks. Defense-GAN [25] use the generator at test time to project the corrupted input on the manifold of the natural examples. Lee et al. [26] introduce Generative Adversarial Trainer (GAT) in which the generator is trained to attack the classifier. Like Adversarial Training [7], GAT requires that adversarial noise does not change the label. Compare with defenses based on robust optimization, we do not put any prior constraint on the adversarial attack. To the contrary, we require that adversarial noise for robust classifier should change the "true" label of the input to confuse the critic. Our formulation has three components (the classifier, the critic, and the attack) and is also related to Triple-GAN [27]. But, in our work: 1) the generator also fools the classifier; 2) we use the implicit dependency between the model and the attack to improve the robustness of the classifier. Also, we use a fixed algorithm to attack the classifier.

## 3 Robust Optimization

We first recall a mathematical formulation for the robust multiclass classification. Let $f(\boldsymbol{x}; \mathbf{W})$ be a $k$-class classifier, e.g. neural network, where $\boldsymbol{x} \in \mathcal{R}^N$ is in the input space and $\mathbf{W}$ are the classifier parameters. The prediction rule is $\hat{k}(\boldsymbol{x}) = \arg\max f(\boldsymbol{x})$. Robust optimization seeks a solution robust to the worst-case input perturbations:

$$\min_{\mathbf{W}} \max_{\boldsymbol{r}_i \in \mathcal{U}_i} \sum_{i=1}^{N} \mathcal{L}(f(\boldsymbol{x}_i + \boldsymbol{r}_i), y_i) \tag{1}$$

where $\mathcal{L}$ is a training loss, $\boldsymbol{r}_i$ is an arbitrary (even adversarial) perturbation for the input $\boldsymbol{x}_i$, and $\mathcal{U}_i$ is an uncertainty set, e.g. $l_p$-norm $\epsilon$-ball $\mathcal{U}_i = \{\boldsymbol{r}_i : \|\boldsymbol{r}_i\|_p \leq \epsilon\}$. Prior information about the task can be used to select a problem-specific uncertainty set $\mathcal{U}$.

Several regularization methods can be shown to be equivalent to the robust optimization, e.g. $l_1$ lasso regression [28] and $l_2$ support vector machine [29]. Adversarial training [7] is a popular regularization method to improve neural network robustness. AT assumes that adversarial noise is label non-changing and trains neural network on the mixture of original and adversarial images:

$$\min_{\mathbf{W}} \sum_{i=1}^{N} \mathcal{L}(f(\boldsymbol{x}_i), y_i) + \lambda \mathcal{L}(f(\boldsymbol{x}_i + \boldsymbol{r}_i), y_i) \tag{2}$$

where $\boldsymbol{r}_i$ is the adversarial perturbation generated using Fast Gradient Sign method (FGSM). Shaham et al. [30] show that adversarial training is a form of robust optimization with $l_\infty$-norm constraint. Madry et al. [17] experimentally argue that Projected Gradient Descent (PGD) adversary is inner maximizer of eq. (1) and, thus, PGD is the optimal first-order attack. Adversarial training with PGD attack increases the robustness of the regularized models compare to the original defense. Margin maximization [19] is another regularization method which generalizes SVM objective to deep neural networks, and, like SVM, it is equivalent to the robust optimization with the margin loss.

Selecting a good uncertainty set $\mathcal{U}$ for robust optimization is crucial. Poorly chosen uncertainty set may result in an overly conservative robust model. Most importantly, each perturbation $\boldsymbol{r} \in \mathcal{U}$ should leave the "true" class of the original input $\boldsymbol{x}$ unchanged. To ensure that the changes of the network prediction are indeed fooling examples, Goodfellow et al. [7] argue in favor of a max-norm perturbation constraint for image classification problems. However, simple disturbance models (e.g. $l_2$- and $l_\infty$-norm $\epsilon$-ball used in adversarial training) are inadequate in practice because the distance to the decision boundary for different examples may significantly vary. To adapt uncertainty set to the problem at hand, several methods have been developed for constructing data-dependent uncertainty sets using statistical hypothesis tests [31]. In this work, we propose a novel approach for learning a robust classifier which is orthogonal to prior robust optimization methods.

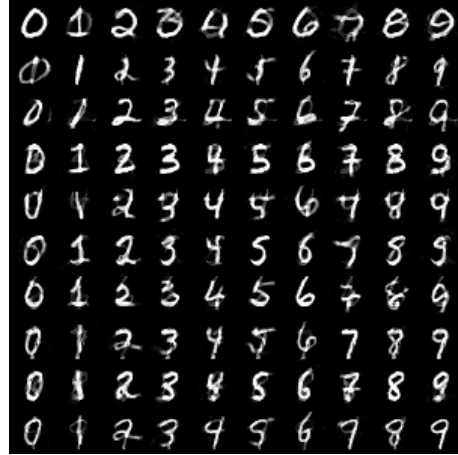

Figure 2: Images off-diagonal are corrupted with the adversarial noise generated by CW [10] $l_2$-norm attack, so the prediction confidence on the adversarial images is at least 95%. The prediction confidence on the original images is 99%.

Ideally, inputs that are adversarial to the classifier should be confusing to a human observer. So the changes introduced by the adversarial noise should be associated with the removing of identifying characteristics of the original label and adding the identifying characteristics of the adversarial target. For example, adversarial images in Figure 2 are visually confusing. The digit '1' (second row, eighth column) after adding the top stroke was classified by the neural network as digit '7'. Likewise, the digit '7' (eighth row, second column) after removing the top stroke was classified by the network as digit '1'. Similarly for other images in Figure 2, the model's "mistakes" can be predicted visually. Such behavior of the classifier is expected and desired for the problems in computer vision. Additionally, it improves the interpretability of the model. In this work, we study image classification problems, but our formulation can be extended to the classification tasks in other domains, e.g. audio or text.

Based on the above intuition, we develop a novel formulation for learning a robust classifier. Classifier is robust if its adversarial examples are indistinguishable from the regular data of the adversarial target (see fig. 1). So, we formulate the following mathematical problem:

$$\min \sum_{i=1}^{N} \mathcal{L}(f(\boldsymbol{x}_i), y_i) + \lambda \mathcal{D}\left[p_{\text{data}}\left(\boldsymbol{x}, y\right), p_{\text{adv}}\left(\boldsymbol{x}, y\right)\right] \tag{3}$$

where $p_{\text{data}}\left(\boldsymbol{x}, y\right)$ and $p_{\text{adv}}\left(\boldsymbol{x}, y\right)$ is the distribution of the natural and the adversarial for $f$ examples and the parameter $\lambda$ controls the trade-off between accuracy and robustness. Note that the distribution $p_{\text{adv}}\left(\boldsymbol{x}, y\right)$ is constructed by transforming natural samples $(\boldsymbol{x}, y) \sim p_{\text{data}}\left(\boldsymbol{x}, y\right)$ with $y \neq y_{\text{adv}}$, so that adversarial example $\boldsymbol{x}_{\text{adv}} = \mathcal{A}_f\left(\boldsymbol{x}; y_{\text{adv}}\right)$ is classified by $f$ as the attack's target $y_{\text{adv}}$.

The first loss in eq. (3), e.g. NLL, fits the model predictive distribution to the data distribution. The second term measures the probabilistic distance between the distribution of the regular and adversarial images and constrains the classifier, so its adversarial examples are indistinguishable from the regular inputs. It is important to note that we minimize a probabilistic distance between joint distributions because the distance between marginal distributions $p_{\text{data}}(\boldsymbol{x})$ and $p_{\text{adv}}(\boldsymbol{x})$ is trivially minimized when $\boldsymbol{r} \sim 0$. Compare with adversarial training, the proposed formulation does not impose the assumption that adversarial noise is label non-changing. To the contrary, we require that adversarial noise for the robust classifier should be visually confusing and, thus, it should change the underlying label of the input. Next, we will describe the implementation details of the proposed defense.

# 4 Robust Learning with Adversary Critic

As we have argued in the previous section, adversarial examples for the robust classifier should be indistinguishable from the regular data of the adversarial target. Minimizing the statistical distance between $p_{\text{data}}(\boldsymbol{x}, y)$ and $p_{\text{adv}}(\boldsymbol{x}, y)$ in eq. (3) requires probability density estimation which in itself is a difficult problem. Instead, we adopt the framework of Generative Adversarial Networks [11]. We rely on a discriminator, or *an adversary critic*, to estimate a measure of difference between two distributions. The discriminator given an input-label pair $(\boldsymbol{x}, y)$ classifies it as either natural or adversarial. For the $k$-class classifier $f$, we implement the adversary critic as a $k$-output neural network (see fig. 3). The objective for the $k$-th output of the discriminator $D$ is to correctly distinguish between natural and adversarial examples of the class $y_k$:

$$\mathcal{L}(f^*, D_k) = \min_{D_k} \mathbb{E}_{\boldsymbol{x} \sim p_{\text{data}}(\boldsymbol{x}|y_k)} \left[ \log D_k(\boldsymbol{x}) \right] + \mathbb{E}_{y:y \neq y_k} \mathbb{E}_{\boldsymbol{x} \sim p_{\text{data}}(\boldsymbol{x}|y)} \left[ \log(1 - D_k(\mathcal{A}_{f^*}(\boldsymbol{x}; y_k))) \right] \tag{4}$$

where $\mathcal{A}_f(\boldsymbol{x}, y_k)$ is the targeted adversarial attack on the classifier $f$ which transforms the input $\boldsymbol{x}$ to the adversarial target $y_k$. An example of such attack is Projected Gradient Descent [12] which iteratively takes a step in the direction of the target $y_k$. Note that the second term in eq. (4) is computed by transforming the regular inputs $(\boldsymbol{x}, y) \sim p_{\text{data}}(\boldsymbol{x}, y)$ with the original label $y$ different from the adversarial target $y_k$.

Our architecture for the discriminator in Figure 3 is slightly different from the previous work on joint distribution matching [27] where the label information was added as the input to each layer of the discriminator. We use class label only in the final classification layer of the discriminator. In the experiments, we observe that with the proposed architecture: 1) the discriminator is more stable during training; 2) the classifier $f$ converges faster and is more robust. We also regularize the adversary critic with a gradient norm penalty [32]. For the gradient norm penalty, we do not interpolate between clean and adversarial images but simply compute the penalty at the real and adversarial data separately. Interestingly, regularizing the gradient of the binary classifier has the interpretation of maximizing the geometric margin [19].

The objective for the classifier $f$ is to minimize the number of mistakes subject to that its adversarial examples generated by the attack $\mathcal{A}_f$ fool the adversary critic $D$:

$$\mathcal{L}(f, D^*) = \min_f \mathbb{E}_{\boldsymbol{x}, y \sim p_{\text{data}}(x, y)} \mathcal{L}(f(\boldsymbol{x}), y) + \lambda \sum_{y_k} \mathbb{E}_{y:y \neq y_k} \mathbb{E}_{\boldsymbol{x} \sim p_{\text{data}}(\boldsymbol{x}|y)} \left[ \log D_k^*(\mathcal{A}_f(\boldsymbol{x}; y_k)) \right] \tag{5}$$

where $\mathcal{L}$ is a standard supervised loss such as negative log-likelihood (NLL) and the parameter $\lambda$ controls the trade-off between test accuracy and classifier robustness. To improve stability of the adversarial mapping during training, we introduce adversarial cycle-consistency constraint which ensures that adversarial mapping $\mathcal{A}_f$ of the adversarial examples should be close to the original:

$$\mathcal{L}_{\text{cycle}}(y_s, y_t) = \mathbb{E}_{\boldsymbol{x} \sim p_{\text{data}}(\boldsymbol{x}|y_s)} \left[ \| \mathcal{A}_f(\mathcal{A}_f(\boldsymbol{x}, y_t), y_s) - \boldsymbol{x} \|_2 \right] \quad \forall y_s \neq y_t \tag{6}$$

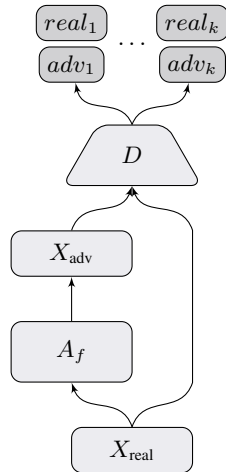

Figure 3: Multiclass Adversary Critic.

**Algorithm 1** High-Confidence Attack $\mathcal{A}_f$

---
1: **Input:** Image $\boldsymbol{x}$, target $y$, network $f$, confidence $C$.
2: **Output:** Adversarial image $\hat{\boldsymbol{x}}$.
3: $\hat{\boldsymbol{x}} \leftarrow \boldsymbol{x}$
4: **while** $p_y(\hat{x}) < C$ **do**
5: $\quad f \leftarrow \log C - \log p_y(\hat{\boldsymbol{x}})$
6: $\quad \boldsymbol{w} \leftarrow \nabla \log p_y(\hat{\boldsymbol{x}})$
7: $\quad \boldsymbol{r} \leftarrow \frac{f}{\|\boldsymbol{w}\|_2^2} \boldsymbol{w}$
8: $\quad \hat{\boldsymbol{x}} \leftarrow \hat{\boldsymbol{x}} + \boldsymbol{r}$
9: **end while**

---

where $y_s$ is the original label of the input and $y_t$ is the adversarial target. Adversarial cycle-consistency constraint is similar to cycle-consistency constraint which was introduced for image-to-image translation [33]. But, we introduce it to constraint the adversarial mapping $\mathcal{A}_f$ and it improves the robustness of the classifier $f$. Next, we discuss implementation of our targeted adversarial attack $\mathcal{A}_f$.

Our defense requires that the adversarial attack $\mathcal{A}_f$ is differentiable. Additionally, adversarial examples generated by the attack $\mathcal{A}_f$ should be misclassified by the network $f$ with high confidence. Adversarial examples which are close to the decision boundary are likely to retain some identifying characteristics of the original class. An attack which optimizes for the mistakes, e.g. DeepFool [13], guarantees the confidence of $\frac{1}{k}$ for $k$-way classifier. To generate high-confidence adversarial examples, we propose a novel adversarial attack which iteratively maximizes the confidence of the adversarial target. The confidence of the target $k$ after adding perturbation $r$ is $p_k(\boldsymbol{x} + \boldsymbol{r})$. The goal of the attack is to find the perturbation, so the adversarial input is misclassified as $k$ with the confidence at least $C$:

$$\min \|\boldsymbol{r}\|$$
$$\text{s.t. } p_k(\boldsymbol{x} + \boldsymbol{r}) \geq C$$

We apply a first-order approximation to the constraint inequality:

$$\min \|\boldsymbol{r}\|$$
$$\text{s.t. } p_k(\boldsymbol{x}) + r\nabla_{\boldsymbol{x}} p_k(\boldsymbol{x}) \geq C$$

Softmax in the final classification layer saturates quickly and shatters the gradient. To avoid small gradients, we use log-likelihood instead. Finally, the $l_2$-norm minimal perturbation can be computed using a method of Lagrange multipliers as follows:

$$\boldsymbol{r}_k = \frac{\log C - \log p_k(\boldsymbol{x})}{\|\nabla_{\boldsymbol{x}} \log p_k(\boldsymbol{x})\|_2} \tag{7}$$

Because we use the approximation of the non-convex decision boundary, we iteratively update perturbation $\boldsymbol{r}$ for $N_{\max}$ steps using eq. (7) until the adversarial input $\boldsymbol{x}_{\mathrm{adv}}$ is misclassified as the target $k$ with the confidence $C$. Our attack can be equivalently written as $\boldsymbol{x}_{\mathrm{adv}} = \boldsymbol{x} + \prod_{i=1}^{N_{\max}} I(p(\boldsymbol{x} + \sum_{j=1}^{i} \boldsymbol{r}_j) \leq C)\boldsymbol{r}_i$ where $I$ is an indicator function. The discrete stopping condition introduces a non-differentiable path in the computational graph. We replace the gradient of the indicator function $I$ with sigmoid-adjusted straight-through estimator during backpropagation [34]. This is a biased estimator but it has low variance and performs well in the experiments.

The proposed attack is similar to Basic Iterative Method (BIM) [12]. BIM takes a fixed $\epsilon$-norm step in the direction of the attack target while our method uses an adaptive step $\gamma = \frac{|\log C - \log p_y(\hat{\boldsymbol{x}})|}{\|\nabla_x \log p_y(\hat{\boldsymbol{x}})\|}$. The difference is important for our defense:

1. BIM introduces an additional parameter $\epsilon$. If $\epsilon$ is too large, then the attack will not be accurate. If $\epsilon$ is too small, then the attack will require many iterations to converge.
2. Both attacks are differentiable. However, for BIM attack during backpropagation, all the gradients $\frac{\partial \boldsymbol{r}_i}{\partial \boldsymbol{w}}$ have an equal weight $\epsilon$. For our attack, the gradients will be weighted adaptively depending on the distance $\gamma$ to the attack's target. The step $\gamma$ for our attack is also fully-differentiable.

Full listing of our attack is shown in algorithm 1. Next, we discuss how we select the adversarial target $y_t$ and the attack's target confidence $C$ during training.

The classifier $f$ approximately characterizes a conditional distribution $p(y|\boldsymbol{x})$. If the classifier $f^*$ is optimal and robust, its adversarial examples generated by the attack $\mathcal{A}_f$ should fool the adversary critic $D$. Therefore, the attack $\mathcal{A}_f$ to fool the critic $D$ should generate adversarial examples with the confidence $C$ equal to the confidence of the classifier $f$ on the regular examples. During training, we maintain a running mean of the confidence score for each class on the regular data. The attack target $y_t$ for the input $\boldsymbol{x}$ with the label $y_s$ can be sampled from the masked uniform distribution. Alternatively, the class with the closest decision boundary [13] can be selected. The latter formulation resulted in more robust classifier $f$ and we used it in all our experiments. This is similar to support vector machine formulation which maximizes the minimum margin.

Finally, we train the classifier $f$ and the adversary critic $D$ jointly using stochastic gradient descent by alternating minimization of Equations (4) and (5). Our formulation has three components (the classifier $f$, the critic $D$, and the attack $\mathcal{A}_f$) and it is similar to Triple-GAN [27] but the generator in our formulation also fools the classifier.

# 5   Experiments

Adversarial training [7] discards the dependency between the model parameters and the adversarial noise. In this work, it is necessary to retain the implicit dependency between the classifier $f$ and the adversarial noise, so we can backpropagate through the adversarial attack $\mathcal{A}_f$. For these reasons, all experiments were conducted using Tensorflow [35] which supports symbolic differentiation and computation on GPU. Backpropagation through our attack requires second-order gradients $\frac{\partial^2 f(\boldsymbol{x};\boldsymbol{w})}{\partial x \partial w}$ which increases computational complexity of our defense. At the same time, this allows the model to anticipate the changes in the adversary and, as we show, significantly improves the model robustness both numerically and perceptually.

We perform experiments on MNIST dataset. While MNIST is a simple classification task, it remains unsolved in the context of robust learning. We evaluate robustness of the models against $l_2$ attacks. Minimal adversarial perturbation $\boldsymbol{r}$ is estimated using DeepFool [13], Carlini and Wagner [10], and the proposed attack. To improve the accuracy of DeepFool and our attack during testing, we clip the $l_2$-norm of perturbation at each iteration to 0.1. Note that our attack with the fixed step is equivalent to Basic Iterative Method [12]. We set the maximum number of iterations for DeepFool and our attack to 500. The target confidence $C$ for our attack is set to the prediction confidence on the original input $\boldsymbol{x}$. DeepFool and our attack do not handle domain constraints explicitly, so we project the perturbation after each update. For Carlini and Wagner [10], we use implementation provided by the authors with default settings for the attack but we reduce the number of optimization iterations from 10000 to 1000. As suggested in [13], we measure the robustness of the model as follows:

$$\rho_{\text{adv}}(\mathcal{A}_f) = \frac{1}{|\mathcal{D}|} \sum_{\mathbf{x} \in \mathcal{D}} \frac{\|\mathbf{r}(\mathbf{x})\|_2}{\|\mathbf{x}\|_2} \tag{8}$$

where $\mathcal{A}_f$ is the attack on the classifier $f$ and $\mathcal{D}$ is the test set.

We compare our defense with reference (no defense), Adversarial Training [7, 8] ($\epsilon = 0.1$), Virtual Adversarial Training (VAT) [16] ($\epsilon = 2.0$), and $l_2$-norm Margin Maximization [19] ($\lambda = 0.1$) defense. We study the robustness of two networks with rectified activation: 1) a fully-connected neural network with three hidden layers of size 1200 units each; 2) Lenet-5 convolutional neural network. We train both networks using Adam optimizer [36] with batch size 100 for 100 epochs. Next, we will describe the training details for our defense.

Our critic has two layers with 1200 units each and leaky rectified activation. We also add Gaussian noise to the input of each layer. We train both the classifier and the critic using Adam [36] with the momentum $\beta_1 = 0.5$. The starting learning rate is set to $5 \cdot 10^{-4}$ and $10^{-3}$ for the classifier and the discriminator respectively. We train our defense for 100 epochs and the learning rate is halved every 40 epochs. We set $\lambda = 0.5$ for fully-connected network and $\lambda = 0.1$ for Lenet-5 network which we selected using validation dataset. Both networks are trained with $\lambda_{\text{rec}} = 10^{-2}$ for the adversarial cycle-consistency loss and $\lambda_{\text{grad}} = 10.0$ for the gradient norm penalty. The number of iterations for our attack $\mathcal{A}_f$ is set to 5. The attack confidence $C$ is set to the running mean class confidence of the classifier on natural images. We pretrain the classifier $f$ for 1 epoch without any regularization to get an initial estimate of the class confidence scores.

Our results for 10 independent runs are summarized in Table 1, where the second column shows the test error on the clean images, and the subsequent columns compare the robustness $\rho$ to DeepFool [13], Carlini and Wagner [10], and our attacks. Our defense significantly increases the robustness of the

| Defense | % | [13] | [10] | Our | Defense | % | [13] | [10] | Our |
|---------|------|-------|-------|-------|---------|------|-------|-------|-------|
| Reference | 1.46 | 0.131 | 0.124 | 0.173 | Reference | 0.64 | 0.157 | 0.148 | 0.207 |
| [7] | 0.90 | 0.228 | 0.210 | 0.299 | [7] | 0.55 | 0.215 | 0.191 | 0.286 |
| [16] | 0.84 | 0.244 | 0.215 | 0.355 | [16] | 0.60 | 0.225 | 0.195 | 0.330 |
| [19] | 0.84 | 0.262 | 0.230 | 0.453 | [19] | 0.54 | 0.248 | 0.225 | 0.470 |
| Our | 1.18 | **0.290** | **0.272** | **0.575** | Our | 0.93 | **0.288** | **0.278** | **0.590** |
| | | (a) | | | | | (b) | | |

Table 1: Results on MNIST dataset for fully-connected network in table 1a and for Lenet-5 convolutional network in table 1b. Column 1: test error on original images. Column 3-5: robustness $\rho$ under DeepFool [13], Carlini and Wagner [10], and the proposed attack.

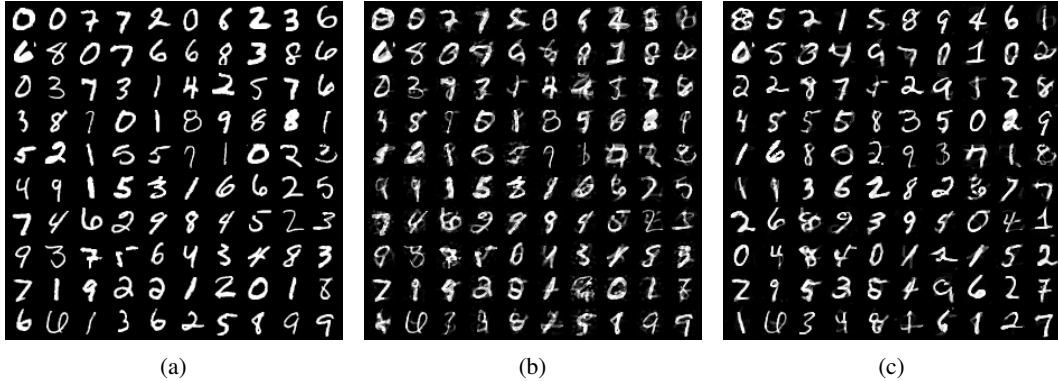

|  | (a) | (b) | (c) |

Figure 4: Figure 4a shows a random subset of test images (average confidence 97%). Figure 4b shows adversarial examples at the class decision boundary (average confidence 34%). Figure 4c shows high-confidence adversarial images (average confidence 98%).

| Defense | % Change | % No change |
| --- | --- | --- |
| Reference | 0.57 | 98.74 |
| [7] | 19.02 | 77.21 |
| [16] | 35.08 | 59.68 |
| [19] | 60.47 | 34.52 |
| Our | 87.99 | 9.86 |

| Defense | % Change | % No change |
| --- | --- | --- |
| Reference | 2.54 | 96.53 |
| [7] | 19.1 | 75.94 |
| [16] | 26.8 | 67.73 |
| [19] | 81.77 | 13.15 |
| Our | 92.29 | 6.51 |

|  | (a) |  | (b) |

Table 2: Results of Amazon Mechanical Turk experiment for fully-connected network in table 2a and for Lenet-5 convolutional network in fig. 4c. Column 2: shows percent of adversarial images which human annotator label with its adversarial target, so adversarial noise changed the "true" label of the input. Column 3: shows percent of the adversarial images which human annotator label with its original label, so adversarial noise did not change the underlying label of the input.

model to adversarial examples. Some adversarial images for the neural network trained with our defense are shown in Figure 4. Adversarial examples are generated using Carlini and Wagner [10] attack with default parameters. As we can observe, adversarial examples at the decision boundary in Figure 4b are visually confusing. At the same time, high-confidence adversarial examples in Figure 4c closely resemble natural images of the adversarial target. We propose to investigate and compare various defenses based on how many of its adversarial "mistakes" are actual mistakes.

We conduct an experiment with human annotators on MTurk. We asked the workers to label adversarial examples. Adversarial examples were generated from the test set using the proposed attack. The attack's target was set to class closest to the decision boundary and the target confidence was set to the model's confidence on the original examples. We split 10000 test images into 400 assignments. Each assignment was completed by one unique annotator. We report the results for four defenses in Table 2. For the model trained without any defense, adversarial noise does not change the label of the input. When the model is trained with our defense, the high-confidence adversarial noise actually changes the label of the input.

# 6 Conclusion

In this paper, we introduce a novel approach for learning a robust classifier. Our defense is based on the intuition that adversarial examples for the robust classifier should be indistinguishable from the regular data of the adversarial target. We formulate a problem of learning robust classifier in the framework of Generative Adversarial Networks. Unlike prior work based on robust optimization, our method does not put any prior constraints on adversarial noise. Our method surpasses in terms of robustness networks trained with adversarial training. In experiments with human annotators, we also show that adversarial examples for our defense are indeed visually confusing. In the future work, we plan to scale our defense to more complex datasets and apply it to the classification tasks in other domains, such as audio or text.

**Acknowledgments**

This work was carried out at the Rapid-Rich Object Search (ROSE) Lab at Nanyang Technological University (NTU), Singapore. The ROSE Lab is supported by the National Research Foundation, Singapore, and the Infocomm Media Development Authority, Singapore. We thank NVIDIA Corporation for the donation of the GeForce Titan X and GeForce Titan X (Pascal) used in this research. We also thank all the anonymous reviewers for their valuable comments and suggestions.

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
