[Reviews · NeurIPS 2018]

Reviewer 1



This paper proposes an adversarial defense. They train a class dependent discriminator to distinguish between real images and those generated by a standard adversarial attack. The central idea of the paper is a good one and the visual evidence of their approach is compelling. I fully expect that the paper will be a good one, but seems to require another revision. The current submission is lacking in several ways, and overall feels rushed. Please have your submission proof-read for English style and grammar issues. The paper purports to propose a novel adversarial attack in section 4, but as they themselves point out on line 210 it is just the basic iterative attack with an adaptive step. This hardly constitutes a novel contribution, and the paper should not pretend it is. I believe the authors would agree, especially considering their later experiments don't use their own novel attack method but instead report results just for DeepFool / Carlini and Wagner and the basic iterative method. If this paper was really proposing a novel attack method worth mentioning as a novel contribution, then I argue it would have been a necessity to at least have included the method in their own evaluations. Overall, I have some concern about the paper. While I like the idea and the method and the visual evidence is good, they relying on the implicit dependence of the generated adversarial image to the parameters of the current classifier, as they note on line 214. However, the attack they use is an iterative one with a stopping condition. This discrete stopping criterion introduces a non differentiable path in the computation graph. The paper does not address this, and so I suspect the authors did not either realize or address this themselves. Simply ignoring the dependence of the stopping criterion (p(hat x) < C) on the parameters of the classifier amounts to a sort of straight through gradient estimation. This will produced biased gradients. This may very well be adequate in practice, but the paper needs to address the issue and does not. While the visual evidence is rather striking, the numerical results are less impressive. Table 1: the [4] vs [10] element presumably has a typo in 2.021. Table 2 has two tables, while the text nor the caption addresses why there are two tables, and the tables have different numbers. Also, none of these human evaluations have any sort of statistics. I don't know how many humans were asked. I have no sense of whether these numbers are statistically significant.

Reviewer 2



Work description The paper proposes a new method for improving robustness of NNs to adversarial attacks. The work progresses towards distinguishing two types of errors. The first type is misclassification of (adversarial) images that may be in or out of the data manifold but are unlikely to be instances of other classes. The second type is misclassification of images that are likely instances of other classes. The second type of errors comes from the inherent overlap of the data and cannot be avoided (e.g. there are digits that are “in between” the classes and are difficult to classify for humans). The first type is the harmful one, which needs to be addressed. The main idea is to achieve that the net misclassifies a perturbed image only if it looks like the target class and is robust to all other perturbations. The proposed learning method explicitly forces the distribution of misclassified images for a given adversarial class to look like the true data samples of that class. The distance between these distributions leads to the regularizer which is expressed as a GAN pair in which the generator is the algorithm that finds adversarial examples leading to the misclassification with high confidence. The experiments on MNIST compare to SOTA methods and show improved robustness results. A human study is conducted to prove that the remaining adversarial perturbations are indeed confusing to the human (and thus are of the second type). Evaluation I thank the authors for their work. The paper is very clear. It gives an excellent overview of existing methods and how they can be viewed as (approximate) robust optimization methods. The method is still a bit “heuristic” in the sense that the regularizing objectives are appended somewhat arbitrary. However, it is much sounder than many other proposals. The experiments are on a simple dataset but cleanly made and giving a comparison to the baseline method and 3 robust training techniques. It is not clear what is the computational complexity of the method (i.e. are second order gradients involved?). However, the positive aspect is that the test-time model is the standard NN and is thus efficient. In my opinion the paper makes a good contribution and I see no problems. Clarity (minor) L 35-36: this is confusing since after adversarial training is introduced, it suggests training with adversarial examples that should be similar to regular images. This stays confusing in Fig 1 and lines 43-44. In Fig 1. it is not explained what is what. A good formulation when I finally got the idea is in lines 133-135. The design of the high confidence attack is perhaps not well explained. I see that without the high confidence the “generator” would only produce images on the classification boundary between the classes. Then enforcing the generator distribution to be close to the data distribution would actually enforce the model to have fractured boundaries. Instead it is somehow desirable to cover the whole adversarial target class, i.e. to also generate the images in its interior. What is the purpose of cycle consistency (7) is not clear. Discussion The difference between classifier and the critique is not completely clear. Could the critique be used as a classifier? Could the critique and the classifier be the same? It seems that works on supervised learning formulations of generative models would be highly relevant. I.e. when one learns the joint mode of data and labels p(y,x). The objective (3) looks like the pseudo-likelihood learning of such generative model: the first part is the conditional likelihood p(y|x), modeled by the classifier, and the second part is the conditional likelihood p(x|y), modeled by the generator (which is the adversarial remapping of the true distribution). In that sense the method performs generative learning – fitting the joint distribution p(y,x) to the data. Such formulations are known to improve generalization properties with scarce data but achieve lower accuracy when more data is available. Alternatively to (3), is it possible to enforce the constraint that “images that are misclassified by the network may not look like the original class”? Technical In (1) max should be under the sum Minor L78: employ utilize Final comments I believe the rebuttal addresses well all raised questions. I think it is an excellent work both conceptually and practically.

Reviewer 3



In this paper, a new regularization method to improve the robustness of neural network is proposed. The new learning problem is formulated as generative adversarial networks, where the adversary critic is to designed to tell regular and adversarial images, and the classifier is to fool the critic with the adversarial images. The goal of this paper is to make the adversarial examples visually confusing, which seems not the case in the previous adversarial training. The new problem is to minimize the traditional classification loss, plus a new regularization term, which is the distance between the distribution of the regular data and the adversarial examples. The experiments show that in terms of robustness, the proposed method outperforms three existing methods and also a simple baseline on MNIST, with both a fully-connected network and a convolution network. There might be a typo in Table 1 row 3, col 3. From what is presented, the result makes sense. But I am not familiar with this field.